# An Implementation of Inverse Cosine Hardware for Sound Rendering Applications

**DOI:** 10.3390/s23156731

**Published:** 2023-07-27

**Authors:** Jinyoung Lee, Cheong-Ghil Kim, Yeon-Kug Moon, Woo-Chan Park

**Affiliations:** 1Department of Computer Science and Engineering, Sejong University, Seoul 05006, Republic of Korea; jylee@rayman.sejong.ac.kr; 2Department of Computer Science, Namseoul University, Cheonan 31020, Republic of Korea; cgkim@nsu.ac.kr; 3Korea Electronics Technology Institute, Seongnam 13509, Republic of Korea; ykmoon@keti.re.kr

**Keywords:** trigonometric, inverse cosine function, hardware

## Abstract

Sound rendering is the process of determining the sound propagation path from an audio source to a listener and generating 3D sound based on it. This task demands complex calculations, including trigonometric functions. This paper presents hardware-based inverse cosine function calculations using the table method and linear approximation. This approach maintains a high accuracy while limiting hardware size for suitability in sound rendering applications. Consequently, our proposed hardware-based inverse cosine calculation method is a valuable tool for achieving high efficiency and accuracy in 3D sound rendering.

## 1. Introduction

Trigonometric functions are fundamental components in various fields, such as 3D graphics computing, digital signal processing, and communication systems. Implementing these functions through software necessitates the execution of hundreds of general-purpose instructions. Nevertheless, the hardware implementation of trigonometric functions is favored because of its enhanced processing speed and reduced power consumption [1]. Given these benefits, hardware-based implementations of trigonometric functions have garnered significant research attention and interest [2]. Typically, such research endeavors aim to optimize trigonometric functions in multiple aspects, encompassing accuracy, processing speed, memory efficiency, and hardware complexity [3,4,5].

Recently, burgeoning interest in blockchain, metaverse, extended reality (XR), virtual reality (VR), and mixed reality (MR) has spurred research to augment realism and immersion. While much of this research concentrates on visual components, high-quality auditory elements are equally crucial for enhancing immersion in virtual environments or multimedia applications [6].

Sound rendering, a technology that computes various physical properties such as echo, refraction, and reflection alongside sensor-oriented information, is instrumental in generating realistic 3D audio [7], requiring intricate calculations, including trigonometric operations. In immersive environments like virtual reality, the role of sensors is particularly crucial as they help provide an authentic and accurate experience by closely simulating real-world acoustics. In the case of [8,9], the use of sensors for capturing hand movements and controlling virtual objects showcases their significance in enhancing virtual experiences. Meanwhile, ref. [10] highlights the diverse information provided by sensors, further emphasizing their importance in various applications.

Sensors play an essential role in providing accurate data for sound rendering, which demands intricate calculations, including trigonometric operations. In this context, sensors are particularly prevalent in semiconductor devices, where they contribute to real-time data acquisition and overall faster processing. The efficient implementation of such technologies is crucial for sensors, which are widely used in the semiconductor industry. By optimizing trigonometric functions, it is possible to enhance the capabilities of these sensors, leading to faster overall processing and improved performance in a wide range of applications.

Calculations for the inverse cosine function are vital to accurately represent physical properties like diffraction, which significantly bolster realism [11]. Accurately implementing trigonometric functions in hardware presents considerable challenges because of the algorithmic complexity, leading to increased hardware size and implementation difficulty [12]. Approaches based on the coordinate rotation digital computer (CORDIC) algorithm [13,14], linear interpolation [15], and polynomial approximation [16] are used to surmount these issues. Conventionally, these methods use table-based approximation techniques that store pre-calculated values to streamline and expedite trigonometric function processing while maintaining high memory efficiency [17,18,19,20]. By integrating efficient trigonometric function implementations with sensor technology, it is possible to achieve faster overall processing, contributing to the enhancement of realism and immersion in various applications.

In this paper, we present an efficient inverse cosine hardware architecture. Our objective is to achieve hardware implementation with minimal errors and reasonable hardware size by adopting a modified linear interpolation method based on the traditional table-based interpolation technique. We propose a modified linear interpolation method that accounts for the characteristics of the inverse cosine function graph by widening partition sections in domains with small slopes and shortening them in domains with large slopes.

This partitioning approach enables us to maintain low error across the entire domain while preserving a small lookup table (LUT) size. The maximum error value of the proposed inverse cosine hardware is less than 0.005, meeting the targeted error range. The proposed architecture is designed to be compatible with the 24 bit floating-point format used in [21,22] and can be applied with high precision. The proposed architecture has been verified through synthesis for register-transfer level (RTL) and application-specific integrated circuit (ASIC) evaluation.

The remainder of the paper is structured as follows. Section 2 describes the background of the data format adopted in this paper and traditional approaches for implementing trigonometric functions in hardware. Section 3 describes the proposed approximation method and inverse cosine hardware, and Section 4 discusses the experimental environment for hardware verification and error analysis under various conditions. Finally, Section 5 concludes the paper.

## 2. Background

### 2.1. Data Format

Figure 1 illustrates a 24 bit data format composed of a 1 bit sign segment, a 6 bit integer segment, and a 17 bit fractional segment. This is the default data format in ray tracing hardware [21] and sound rendering hardware [22] to reduce design complexity while maintaining a high quality. Importantly, this 24 bit data format is also used as the input for our proposed inverse cosine hardware, being received in a floating-point representation. To keep in alignment with our objective of minimizing logic size, once received, the data format is then converted to a fixed-point representation. Nonetheless, compared with a 32 bit precision, there is potential for diminished accuracy, although it is a trade-off we consciously accepted in our design process to ensure a balance between performance and complexity.

Previous research [21] addressed this accuracy issue by devising designs that minimize precision errors, resulting in the benchmark from [21] demonstrating no discernible visual issues. Consequently, this study also adopts a 24 bit precision data format capable of delivering high-quality outcomes using an appropriate design.

### 2.2. Linear Approximation for Trigonometric Functions

Figure 2 illustrates an example of linear interpolation. The function *y* = *mx* + *b*, derived from linear interpolation in Section 3, can approximate the value of the original function g(x). Thus, an approximation with a minimal error rate can be attained by appropriately partitioning the domain of a given function and using linear interpolation.

An approximation method like linear interpolation for implementing trigonometric functions offers several advantages [23]. First, it exhibits rapid processing speed. The computation speed is significantly enhanced by storing precomputed trigonometric function values in a table and using them for linear interpolation. Second, it consumes less memory. The use of discretized function values for linear interpolation decreases overall memory usage. Consequently, implementing trigonometric functions with linear interpolation is advantageous in systems with limited memory, such as embedded systems.

### 2.3. Look-Up Table

The LUT method has been used extensively for calculating trigonometric functions [5,24,25]. This approach stores pre-calculated trigonometric function values in a table, facilitating rapid access to these values. However, relying solely on the LUT method can produce errors when the input value does not correspond to the discrete values in the table.

The LUT method is frequently combined with linear interpolation to address this issue. Moreover, because the LUT method uses pre-calculated values, incorporating linear interpolation does not considerably affect processing speed. Thus, the use of both linear interpolation and the LUT method enables a high accuracy and swift processing speed. This technique also reduces memory usage and hardware complexity, making it a traditionally favored choice in trigonometric functions.

### 2.4. Applying Linear Interpolation and Lookup Table for the Inverse Cosine Function

Figure 3 illustrates the graph of Equation (1), representing the inverse cosine function divided by PI. The range of the inverse cosine function spans [0, PI], while its domain is [−1, 1]. Several assumptions are made in this study to facilitate hardware implementation. First, the range is adjusted to [0, 1] by dividing the inverse cosine function by PI. Moreover, the output value is considered solely for the domain [0, 1]. Given the graph’s symmetry with respect to the coordinate (0, 1/2), output values for the domain [−1, 0] can be derived by subtracting 1 from the output values for the domain [0, 1]. These assumptions streamline hardware implementation without compromising accuracy.
(1)y=acos(x)/π,
f(x) → y = m_s_x + b_s_, (s = 0, 1, 2, ~, *k*),(2)

Equation (2) defines the lines corresponding to each partition section ‘s’ when the domain of the inverse cosine function is segmented into *k* sections for linear interpolation. Given an arbitrary input value, the equation of the line corresponding to the partition section encompassing the input value is constructed and used as the linear approximation in Equation (1). The equation of a line can be defined using the gradient *m_s_* and bias *b_s_* in Equation (2). Typically, *m_s_* and *b_s_* are pre-calculated and stored in the LUT, accessed based on the input value.

Figure 4 illustrates an example of linear interpolation applied to the domain [0, 1] of the inverse cosine function. The linear approximation error diminishes as the domain is subdivided more precisely (as *k* increases), but the size of the LUT also expands. Consequently, implementing a domain partitioning method optimized for targeted accuracy is essential.

## 3. Proposed Inverse Cosine Hardware

In this section, the methods applied to the proposed hardware are introduced. Section 3.1 describes our design choices, Section 3.2 describes the LUT construction method based on efficient domain partitioning, and Section 3.3 explains the structure and operation of the proposed inverse cosine hardware.

### 3.1. Design Decisions

In this section, we describe some of the underlying goals that have influenced the design decisions of the inverse cosine hardware architecture. These goals encompass processing speed, memory efficiency, hardware complexity, and the minimization of arithmetic units in trigonometric functions.

Processing Speed. Particularly in real-time systems, the temporal efficiency in task completion is of utmost importance, emphasizing the objective of latency minimization. Consequently, an increase in processing speed, achieved through a reduction in latency, stands as a salient metric and a crucial target in the effective optimization and implementation of hardware.

Memory Efficiency. Improving memory efficiency means systems have been optimized to use memory more effectively or to perform the same tasks with less memory. A common element between the CORDIC algorithm and our proposed inverse cosine hardware is the trade-off that exists between the size of the lookup table (LUT) and the accuracy of computations. An enhancement in memory efficiency can be ascribed to a situation where the size of the LUT is diminished while simultaneously maintaining or augmenting the accuracy of the calculations.

Hardware Complexity. The CORDIC algorithm holds a significant advantage of simplicity in hardware implementation due to its reliance on elementary operations such as basic addition, subtraction, and bit shift. However, an increase in accuracy might necessitate the inclusion of complex multiplication or division operations, consequently escalating the complexity of the hardware. This involves optimizing the hardware architecture to be straightforward to implement and be devoid of convoluted operational expressions, all the while not compromising on accuracy.

Minimizing Hardware Complexity. Hardware complexity is a function of various parameters, including the number of operations, circuit size, power requirements, and the intricacy of the overall design. Minimizing such complexity is essential for seamless implementation and rapid operation. Nevertheless, potential limitations in terms of performance or functionality must be considered, thereby emphasizing the significance of striking a balance between hardware complexity and performance. In our manuscript, we present an inverse cosine hardware that endeavors to minimize hardware complexity without forfeiting accuracy.

Minimizing Arithmetic Units in Trigonometric Functions. In the architecture of our proposed inverse cosine hardware, we have made concerted efforts to minimize the number of arithmetic units employed. This design utilizes a LUT reference to yield the result with merely a single multiplication and addition operation. The ability to maintain a high accuracy with such rudimentary operations is perceived as a notable achievement.

### 3.2. Approximation Method for Inverse Cosine Function

The accuracy and size of the proposed inverse cosine hardware are contingent on the total number of partitioned sections (*k*) and the partitioning approach. We propose a domain partitioning method to minimize the size of the LUT while adhering to the desired error range. The proposed method segments the main domain into multiple sub-domains and partitions the sub-domains that do not meet the error range criteria. As the partitioning process iterates, the error diminishes, and the degree of partitioning may vary according to the targeted error range.

Figure 5 illustrates the inverse cosine function and the interval where the slope becomes markedly steep within the inverse cosine function. Figure 5 (right) is a magnification of the domain [0.9, 1] from Figure 5 (left). The magnified graph includes a region where the slope steepens significantly. In particular, within the domain [0.98, 1.00], the slope can be observed to be very steep. The actual values of the gradient can be verified through Table 1.

Table 1 presents the gradient values for sections 1–3, and the last two sections of the partitioned intervals. Four cases were considered for the value of *k*: 64, 128, 256, and 512. In the gentle sections of the inverse cosine graph curve, namely sections 1–3, it can be observed that the gradient values of the linear equations do not change even as the value of *k* increases. On the contrary, in the steep sections of the graph curve, specifically the last part of the divided sections, the gradient values of the linear equations differ significantly depending on the value of *k*. This suggests that the graph curve is indeed becoming sharply steep, and the common method for linear approximation in such cases is to further sub-divide the domain.

Table 2 presents the number of LUT entries for various *k* values, indicating the LUT size. For the inverse cosine, the main domain is the domain [0, 1]. Thus, k_a_ represents the partitioning number of the domain [0, 1], and k_b_ refers to the partitioning number of sub-domains with a sharp slope. In all three examples, the sub-domain division size is set to be the same as when the main domain is divided into 2^13^ sections. We establish the error range as ‘less than 0.00125’, so the *k* value must be 2^13^ or greater. Further explanation on this is provided in Section 4.2 Error Analysis.

Table 2 (a) corresponds to the case where the domain [0, 1] is divided into 2^4^ sections, and the sharp slope section (i.e., the section where errors occur) is further divided into 2^9^ sub-sections. The number of LUT entries is 527. Table 2 (b) corresponds to the case where the domain [0, 1] is divided into 2^5^ sections, and the error section is divided into 2^8^ sub-sections. The number of LUT entries is 287. Table 2 (c) corresponds to the case where the domain [0, 1] is divided into 2^7^ sections, and the error section is divided into 2^6^ sub-sections. The number of LUT entries is 191. All three satisfy the error tolerance we set, but there is a significant difference in the total number of LUT entries.

Figure 6 illustrates the partitioning method used in this study and the calculation process for the reference address of the LUT given an input value. In this instance, the partitioning method with the fewest LUT entries from Table 2 in row (c) was selected. There are two approaches to partitioning the domain for C. The method presented in Figure 6 involves dividing the domain [0, 1] into 2^7^ sections and then dividing the sub-section where errors occur into 2^6^ sections.

Given an input value, the reference address in the LUT can be derived from the upper 13 fraction bits of the input value *s* in Equation (2). When the domain [0, 1] is divided into 2^7^ sections, the reference address for segments 0–127 can be obtained from the upper 7 bits of the input value’s fractional part. For instance, if the upper 7 bits are 0, the input value belongs to Section 0; if the upper 7 bits are 126, the input value belongs to Section 126. Thus, if the upper 7 bits are 127, the input value’s reference address is encompassed within sections 127–190, which are the segments produced by partitioning 2^7^ sections into 2^6^. In this case, the reference address can be determined by adding 127 and the lower 6 bits of the input value’s fractional part.

There are two methods for constructing LUTs: one that generates multiple independent LUTs for each divided section and another that combines them into a single unified LUT. While the method of creating multiple LUTs offers certain advantages in terms of processing speed, the unified single LUT method has benefits in reducing hardware size. Our objective is to satisfy the predetermined error range while minimizing hardware size, so we have opted for the unified single LUT approach.

### 3.3. Proposed Inverse Cosine Function Hardware

Figure 7 illustrates the overall architecture of the proposed inverse cosine hardware, which comprises three stages in a pipelined design. The operational process proceeds as follows. In the first stage, preprocessing is conducted, encompassing underflow/overflow checks, data format conversion, and operations for LUT referencing. In the second stage, the actual computation for the linear approximation in Equation (2) is executed, incorporating a multiplier and an adder module. In the final stage, the output is derived through post-processing, involving data format conversion and underflow/overflow check results.

The lookup tables (LUTs) for the gradient and bias are implemented as read-only memory (ROM) in our proposed inverse cosine hardware. This decision is based on two primary reasons. Firstly, once the partitioning scheme satisfies the target error value, the gradient and bias, which constitute the linear equations for each divided section, are pre-calculated. As these values are constant and do not change, implementing them in ROM is appropriate. Secondly, our hardware is designed with the consideration of being embedded into a sound rendering device. In such an environment, spatial efficiency and power consumption become critical factors, and ROM fulfills these requirements.

The input for the proposed hardware is passed to the fixed-point formatting module and the underflow/overflow module. The fixed-point formatting module converts the input data format from the floating-point to fixed-point and transfers it to the multiplier module. Moreover, the upper 13 bits of the converted fixed-point’s fractional part are passed to the RefAddr module for calculating the reference address (Figure 6). The gradient LUT and bias LUT output ‘m_s_’ and ‘b_s_’ values are based on the calculated reference location from Equation (2). These ‘m_s_’ and ‘b_s_’ values are passed to the multiplier and adder modules.

The multiplier module performs multiplication operations on the inputs received from the fixed-point formatting module and gradient LUT and forwards the result to the adder module. The adder module performs additional operations on the inputs received from the multiplier module and bias LUT and transfers the result to the result formatting module.

The result formatting module outputs 0 if overflow occurs or 1 if underflow occurs based on the signals received from the underflow/overflow check module. If neither underflow nor overflow occurs, the value received from the adder module is converted to a floating-point format and outputted.

## 4. Experiment

### 4.1. Experiment Environment

Figure 8 presents the experimental environment devised for the validation of the proposed inverse cosine hardware. The flow of processing in this environment proceeds as follows. Input data, encompassing all possible data expressible in 24 bit floating-point format (0 × 0~0 × 3E0000) within the range of 0 to 1, are transmitted to software and hardware components.

The software for validating the functionality of the inverse cosine function is comprised of a C model and an emulator version. The C model utilizes the inverse cosine function library provided by the C programming language. On the other hand, although the emulator is based on the C language, it implements the inverse cosine function in a hardware-like manner, reflecting the exact operations of the hardware.

The output values from the C model serve as a reference for the received inputs. Consequently, the emulator’s output values are compared to these reference values to calculate the error. The inverse cosine hardware, built using Verilog HDL based on Figure 7, undergoes validation by comparing its output values with those of the emulator.

Adding to this, the primary purpose of the emulator, designed for the validation of the proposed inverse cosine hardware, is to reflect the operations of the hardware as accurately as possible and verify the accuracy of the output values through this reflection. During this process, the error was calculated by comparing the output values of the emulator with the inverse cosine function available in the standard library of the C language. This approach aims to ensure the accuracy of the hardware while emphasizing the practicality of hardware implementation.

### 4.2. Error Analysis

Section 3.2, “Approximation Method for Inverse Cosine Function,” discusses setting the range for the predefined error value to be less than 0.00125. The inverse cosine hardware we propose is designed to incorporate the diffraction characteristics [26] of sound rendering hardware. The most crucial aspect of implementing the diffraction function is accurately determining the diffraction path from the source to the listener. The set error range was initially sufficiently small to prevent errors in identifying diffraction paths. However, we identified corner cases where the error exceeds this range as we approach one, but we have determined that this level of error does not interfere with the function of the sound rendering hardware. As a result, we have adjusted the final error range to be less than 0.005. This range can still be fine-tuned according to specific requirements or circumstances.

Table 3 presents the F_Input, maximum error value (MEV), and reduction rate of MEV for various *k* values. F_Input is the input value where the error initially occurs, and MEV is the maximum error value. The reduction rate of MEV is the rate of decrease in MEV as *k* doubles based on the MEV value at *k* = 64.

As *k* doubles, the MEV value declines by approximately 70%. Based on this rate of decrease, to meet our permissible initial error range (less than 0.00125), *k* must be at least 2^13^, implying that the domain must be divided into 8192 sections. However, partitioning the entire range into 8192 sections is inefficient as it necessitates a significantly large LUT size.

Lower *k* values can be applied to domains with a gentle slope. Two partitioning methods can be considered to minimize LUT size while satisfying the allowable error range under the condition of dividing the domain into at least 2^13^ sections. The first method is to divide the domain [0, 1] into 2^6^ sections and subdivide the last section (including x-coordinate 1) into 2^7^ sections. The second method is to divide the domain [0, 1] into 2^7^ sections and then partition the last section into 2^6^ sections.

Both methods are identical in that F_Input is included in the last section, and two LUTs (gradient, bias LUT) have 191 entries. Consequently, regardless of the chosen method, the size and error remain the same. In this study, we adopted the latter method for implementation, and the number of LUT entries can be found in Table 1. The partitioning method we used is illustrated in Figure 6.

### 4.3. ASIC Evaluation

Table 4 presents the synthesis results and resource utilization for the proposed inverse cosine hardware. The Xilinx Alveo U250 Data Center accelerator card was used as the synthesis target board, and the Vivado tool was used for synthesis.

For ASIC evaluation, 28-nanometer low-power process technology and a Synopsys Design Compiler were used. The inverse cosine hardware was synthesized at a clock period of 1.3 ns, corresponding to an operational frequency of 769 MHz. Table 5 presents the ASIC evaluation results, including the global cell area and gate count. The total area of the proposed hardware is approximately 2672 μm^2^, and the gate count is estimated at 7342.

## 5. Conclusions

In this paper, we proposed an efficient inverse cosine hardware architecture for sound rendering hardware. We achieve a small hardware size with a minimal error by applying a modified linear interpolation method based on the traditional table-based approach. The proposed architecture is compatible with the 24 bit floating-point format used in previous research and can also be applied to obtain a high precision. Furthermore, the inverse cosine hardware was validated through RTL simulation and ASIC evaluation, and the experimental results confirmed high accuracy and efficiency.

Our approach, based on the modified linear interpolation method, has achieved high accuracy within a reasonable hardware size. By offering rapid processing through table lookup instead of complex calculations, our proposed inverse cosine hardware, which has been designed considering logic size and latency, is expected to contribute to enhancing efficiency when embedded in resource-limited applications. The inverse cosine hardware proposed in this paper is already being used in real-time for sound energy calculations in sound rendering, dealing with various physical phenomena such as diffraction. Notably, it has demonstrated that it can perform real-time operations without issue even in dynamic scenes with numerous sound sources, handling up to 16 sources simultaneously.

In the future, there will be a need for more diverse research on bit precision and error tolerance. Furthermore, while this study focused only on the inverse cosine function, there is a need for efficient hardware architecture research on other mathematical functions as well. Through this, it can develop into a more practical technology and provide improved performance in sound rendering and other multimedia applications.

## Figures and Tables

**Figure 1 sensors-23-06731-f001:**
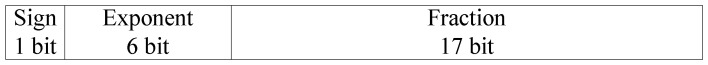
A 24 bit data format. This data format, in floating-point representation, is used as the input for the inverse cosine hardware. After being received, the data format is converted to a fixed-point representation, considering the minimization of logic size.

**Figure 2 sensors-23-06731-f002:**
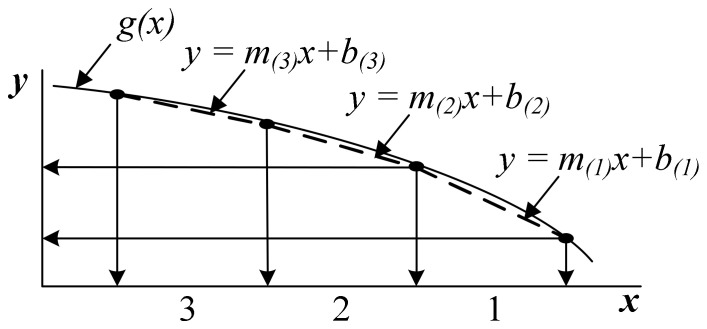
An example of linear interpolation. The function g(x) is divided into three sections, each representing the form of a linear equation.

**Figure 3 sensors-23-06731-f003:**
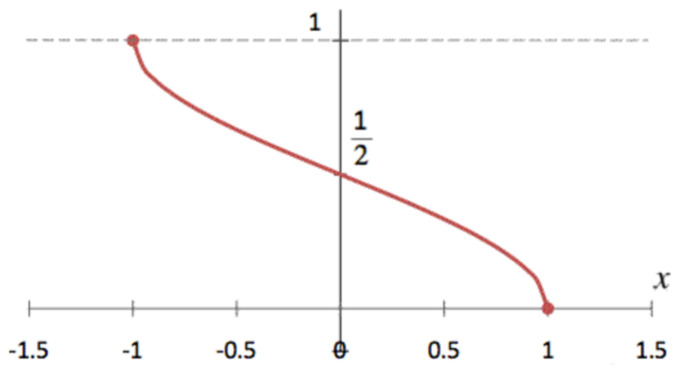
The graph of the inverse cosine function divided by pi. It is noticeable that the range of the original inverse cosine function, [0, pi], has been transformed to the range [0, 1].

**Figure 4 sensors-23-06731-f004:**
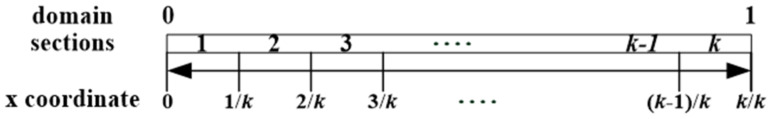
A schematic of dividing sections for the domain [0, 1].

**Figure 5 sensors-23-06731-f005:**
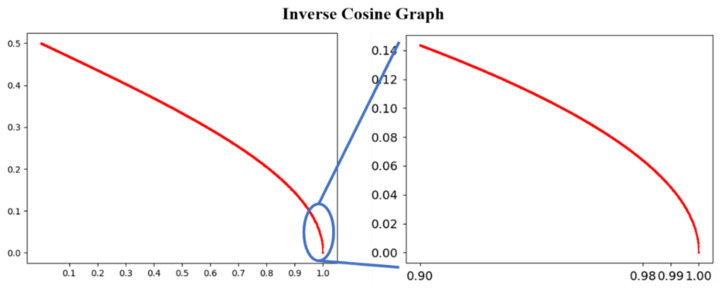
(**Left**) Inverse cosine graph, and (**Right**) the inverse cosine graph in the section of the domain [0.9, 1.00]; it can be observed that the slope dramatically increases as it surpasses approximately 0.99.

**Figure 6 sensors-23-06731-f006:**
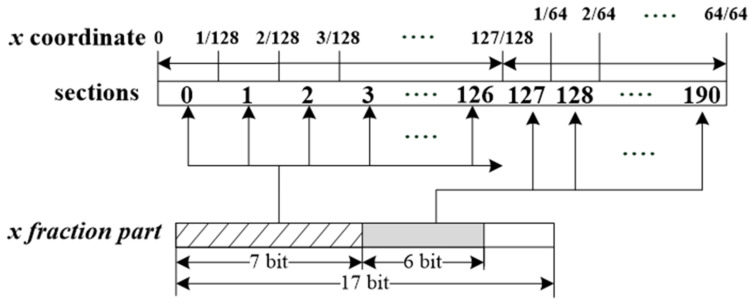
Schematic of dividing sections for domain [0, 1]. The sections corresponding to the main domain range from the 0th to the 128th x-coordinates. The sections that pertain to the sub-domain are included within the 128th x-coordinate, which are divided into 64 parts.

**Figure 7 sensors-23-06731-f007:**
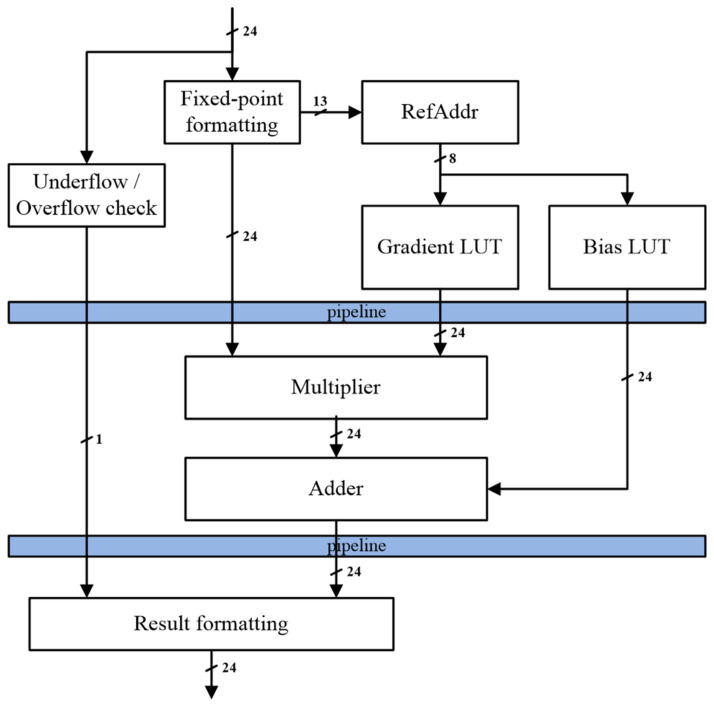
Block diagram of proposed inverse cosine hardware architecture.

**Figure 8 sensors-23-06731-f008:**
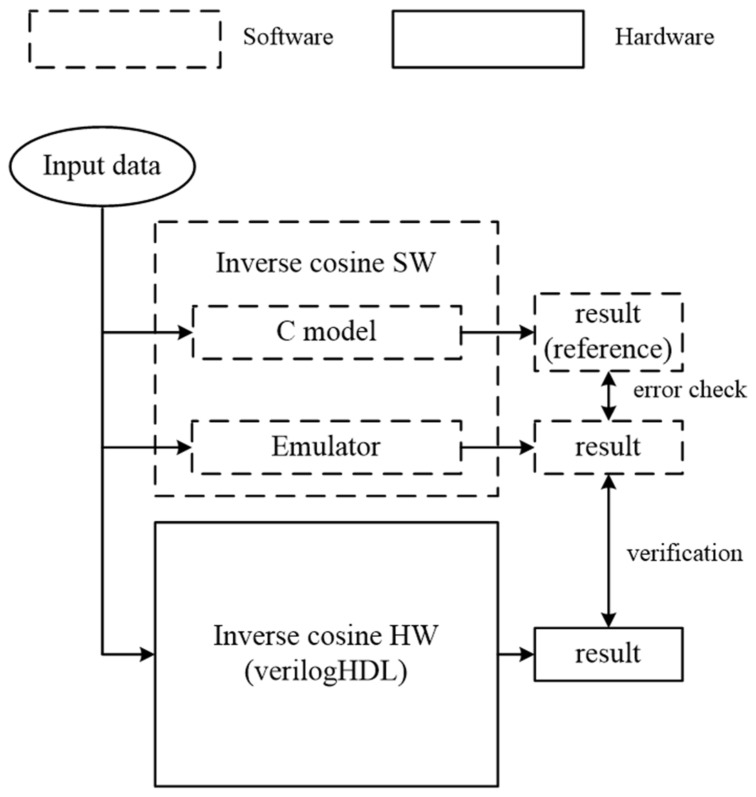
An experimental environment for validation. It was composed of a software model based on the C programming language and a hardware model based on Verilog HDL.

**Table 1 sensors-23-06731-t001:** The gradient values of linear equations for each section divided by *k* in the domain [0, 1] of the inverse cosine graph.

	Divided Sections					
*k*	1	2	3	-	Second to Last Section	Last Section
	Gradient					
64	−0.31	−0.31	−0.31	-	−1.50	−3.60
128	−0.31	−0.31	−0.31	-	−2.11	−5.09
256	−0.31	−0.31	−0.31	-	−2.98	−7.20
512	−0.31	−0.31	−0.31	-	−4.22	−10.18

**Table 2 sensors-23-06731-t002:** Number of LUT entries according to *k* values.

	k_a_ for Domain [0, 1] (Main Domain)	k_b_ for Section with Sharp Slope (Sub-Domain)	Total Entries
(a)	2^4^	2^9^	527
(b)	2^5^	2^8^	287
(c)	2^7^ (2^6^)	2^6^ (2^7^)	191

**Table 3 sensors-23-06731-t003:** Error analysis based on *k*.

*k*	64	128	256	512
F_input	0.9850	0.9926	0.9964	0.9983
Maximum Error Value	0.01405	0.0099	0.0070	0.0049
Reduction Rate of MEV	Standard	70.764	70.747	70.659

**Table 4 sensors-23-06731-t004:** Resource utilization of proposed inverse cosine hardware.

	LUT	REGISTER	BLOCK MEMORY	URAM
Used	224	58	0.5	0
Available	1,728,000	3,456,000	2688	1280
Utilization (%)	0.01	<0.01	0.02	0

**Table 5 sensors-23-06731-t005:** Area results according to the ASIC evaluation.

	Total Area (μm^2^)	Gate Counts	Operation Frequency
Inverse Cosine Hardware	2672.7	7342	769 Mhz

## Data Availability

Not applicable.

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
