# Peer review of "An Implementation of Inverse Cosine Hardware for Sound Rendering Applications"

_sensors, 2023, doi:10.3390/s23156731_

Round 1

Reviewer 1 Report

This paper present an architecture for inverse cosine function calculation. The maximum error value of the proposed hardware is less than 0.00125, meeting the targeted error range.

I am not sure whether the paper should be publish in Sensor journal as the paper covers mostly hardware implementation. There is very little description of the whole sensors system in order to justify the selected journal submission.

Another drawback of the paper is a very little discussion on the selected architecture, e.g. why CORDIC implementation is not implemented, in my opinion the CORDIC architecture might be a better solution in the case when 10 clock cycles pipeline latency is allowed. Does the latency matters in the described system?

In this paper it should be clearly stated what kind of operations are used: fixed or floating point. I assume that the input is a 24-bit floating point however the maximum allowed error / rounding does not comply with the floating point standard.

Summing up, the proposed paper solves a very specific engineering problem, but I am not convinced that the proposed solution is an optimal one. However I have not found a similar papers covering a the problem. In my opinion, some more literature study / implementation results comparison is welcome.

Figure 1. 24-bit fixed-point precision.

should be floating point format

106: The use of discretized function values for linear interpolation decreases overall memory usage.

The CORDIC does not use memory at all!

Reviewer 2 Report

The authors present an efficient inverse cosine hardware implementation with reported minimal errors and reasonable hardware size.

It is difficult to assess how this approach improves the realism and immersion in sound rendering unless it is practically validated in one such application. Moreover, while the authors show that their proposed implementation takes few hardware resources, I take it to mean that their method is efficient because it uses less resources.

I don't see any novelty in this work. The modified linear interpolation method does the obvious to meet the error threshold by taking more points in regions of large slopes and less points where slopes do not change much.

Introduction
------------

1). "Such research endeavors aim to optimize trigonometric functions in multiple aspects, encompassing accuracy, processing speed, memory efficiency, and hardware complexity [3,4,5]."

You must define these terms so that the reader understands what is exactly optimized. I understand accuracy. What about processing speed? Is it ops/sec or just sec? What is memory efficiency? What is hardware complexity? How is hardware complexity minimized? Is it no. of arithmetic units used in trigonometric functions minimized?

Section 2.2
-----------

"First, it exhibits rapid processing speed."

You must define this term before you use it. Should I presume this to be number of fixed-point operations per second?

"The use of discretized function values for linear interpolation decreases overall memory usage."

Is memory usage such a big concern? I understand you are talking about embedded systems. How many discrete values one must need to maintain the desired accuracy?

3.1. Approximation Method for Inverse Cosine Function
-----------------------------------------------------

First, you should have a formal partitioning algorithm for determining the best values of Ka and Kb. Then you can explain it informally in the text as you have done.

In its present form, this section is really difficult to understand. Maybe, you can illustrate which regions in the inverse cosine have sharp slope and which don't. I presume that further partitioning into sub-domains takes place in the sharp slopes to meet the error criterion.

3.2. Proposed Inverse Cosine Function Hardware
-----------------------------------------------------

Where are the LUT entries stored in the cosine hardware architecture? For example, from Table 1, the optimal number of LUT entries are 191.

4. Experimental Results
-----------------------

1). What are the execution times of your emulator compared with the execution time to obtain the C model output?
2). Your ASIC evaluation in 4.3 ought to be compared with what hardware resources are used in state-of-the-art hardware cosine implementations. Otherwise, it is difficult to appreciate the figures.

5. Conclusion
-------------

"Moreover, our method can enhance speed and power efficiency in applications such as sound rendering by offering rapid processing via table lookup instead of complex calculations. We anticipate that our proposed inverse cosine hardware can improve the performance of sound rendering hardware."

Please do not make unsubstantiated statements without thorough experimental evaluation. I do not see any reference to how speed is measured or power is measured.

Some general comments:

1). Try to be more descriptive in the Figure captions. It applies to all figures. Repeat or rewrite the caption if you have to from the description in the main text.

None

Round 2

Reviewer 1 Report

This paper presents an architecture for inverse cosine function calculation. The maximum error value of the proposed hardware is less than 0.00125, meeting the targeted error range but it does not comply with the maximum error limit: 0.5 or 1 ULP – the standard for floating point arithmetic. This should be clearly stated in the paper and was my question in my previous review.

> Memory Efficiency. [..] A common element between the CORDIC algorithm and our proposed inverse cosine hard- 183

ware is the trade-off that exists between the size of the Lookup Table (LUT) and the accu- 184

racy of computations. An enhancement in memory efficiency can be ascribed to a situation 185

where the size of the LUT is diminished while simultaneously maintaining or augmenting 186

the accuracy of the calculations.

I do not understand the above. Do you mean that for the proposed algorithm the accuracy can be met by increasing memory size? For CORDIC algorithms the accuracy can be increased by the larger number of iterations (more hardware resources required and larger latency)?

> However, an increase in accuracy might necessi- 190

tate the inclusion of complex multiplication or division operations, consequently escalat- 191

ing the complexity of the hardware. This involves optimizing the hardware architecture 192

to be straightforward to implement and devoid of convoluted operational expressions, all 193

the while not compromising on accuracy.

I am not sure whether you understand the CORDIC algorithm, especially pipeline version where no memory and no shift operation (the hardwired shift operation does not require any hardware!) is required.

> Minimizing Hardware Complexity (subsection)

Bla bla … it is the language of philosophy, not the science.

Similar subsection: Minimizing Arithmetic Units in Trigonometric Functions

In my opinion Section 3.1. should be completely rewritten. Could you please describe you sensor system, e.g. how many arccos() functions calls are required? Why latency is critical? Can you use pipeline version of your function? Is Cordic (pipeline version) algorithm worse? What about Cordic Radix-4 algorithm to reduce the latency?

Derivative (arccos(x) ) = - 1/SQRT(1 – x^2)

For x approaching 1 the  derivative approaches the infinity. Therefore it is very difficult to use linear approximation for x approaching 1!!! It should be clarified in the paper. The last segment is very critical!

For example: for dx= 2^(-13) and when we take the last segment x from 1-dx to 1 we obtain: b= arccos(1-dx)= 0.015625, m= -arccos(1-dx)= 128.0013. Then for x= 1-dx/2 (in the middle of the last segment) we obtain: LinearAproachimation= 0.007812579, arccos(1-dx/2) = 0.011049, error= 0.00324. This error is larger than declared in the paper: 0.0015.

Have you tried to find out the maximum error in symbolic way assuming that the maximum error is in the last segment?

Author Response

Kindly refer to the attached Word document, please.

Reviewer 2 Report

The authors have addressed my comments and concerns well. I recommend publication of this article.

No comments.

Author Response

Your recommendation for publication is deeply appreciated. Thank you very much.

Round 3

Reviewer 1 Report

No more comments.